# Pharmacological Targeting of KCa Channels to Improve Endothelial Function in the Spontaneously Hypertensive Rat

**DOI:** 10.3390/ijms20143481

**Published:** 2019-07-16

**Authors:** Rayan Khaddaj Mallat, Cini Mathew John, Ramesh C Mishra, Dylan J Kendrick, Andrew P Braun

**Affiliations:** Department of Physiology and Pharmacology and Libin Cardiovascular Institute of Alberta, Cumming School of Medicine, University of Calgary, 3330 Hospital Drive NW, Calgary, AB T2N 4N1, Canada

**Keywords:** SHR, Ca^2+^-activated K^+^ channel, endothelium, resistance artery, blood pressure

## Abstract

Systemic hypertension is a major risk factor for the development of cardiovascular disease and is often associated with endothelial dysfunction. KCa2.3 and KCa3.1 channels are expressed in the vascular endothelium and contribute to stimulus-evoked vasodilation. We hypothesized that acute treatment with SKA-31, a selective activator of KCa2.x and KCa3.1 channels, would improve endothelium-dependent vasodilation and transiently lower mean arterial pressure (MAP) in male, spontaneously hypertensive rats (SHRs). Isolated vascular preparations exhibited impaired vasodilation in response to bradykinin (i.e., endothelial dysfunction) compared with Wistar controls, which was associated with decreased bradykinin receptor expression in mesenteric arteries. In contrast, similar levels of endothelial KCa channel expression were observed, and SKA-31 evoked vasodilation was comparable in vascular preparations from both strains. Addition of a low concentration of SKA-31 (i.e., 0.2–0.3 μM) failed to augment bradykinin-induced vasodilation in arteries from SHRs. However, responses to acetylcholine were enhanced. Surprisingly, acute bolus administration of SKA-31 in vivo (30 mg/kg, i.p. injection) modestly elevated MAP compared with vehicle injection. In summary, pharmacological targeting of endothelial KCa channels in SHRs did not readily reverse endothelial dysfunction in situ, or lower MAP in vivo. SHRs thus appear to be less responsive to endothelial KCa channel activators, which may be related to their vascular pathology.

## 1. Introduction

Hypertension contributes significantly to worldwide cardiovascular morbidity and mortality [1,2,3], and appears to have a complex association with endothelial dysfunction, a phenotypic alteration of the vascular endothelium that precedes and predicts the development of adverse cardiovascular events [4,5,6]. Impaired relaxation of large and small arteries is a hallmark feature of hypertension, and may accelerate the progression of atherosclerosis [7,8], due in part to reduced nitric oxide (NO) bioavailability and the actions of endothelium-derived contracting factors (e.g., endothelin-1, thromboxane A_2_, prostaglandin F_2alpha_) [9,10,11]. Hypertension-related endothelial dysfunction may also compromise endothelium-dependent hyperpolarization (EDH), a vasorelaxant process driven by the agonist-evoked activation of small- and intermediate-conductance, Ca^2+^-activated K^+^ channels (KCa2.3 and KCa3.1, respectively) in the endothelium [12,13]. EDH signaling decreases voltage-gated Ca^2+^ channel activity and cytosolic free Ca^2+^ in smooth muscle, leading to reduced contractility. KCa2.3 and KCa3.1 channel activation contributes directly to endothelium-dependent vasodilation by agonists such as acetylcholine and bradykinin, and this electrical signaling further promotes endothelium-dependent NO synthesis [14,15,16]. Alterations of endothelial ion channels (e.g., KCa2.3 channels) have been implicated in the hypertension-associated impairment of EDH [17]. Moreover, genetic inactivation of endothelial KCa2.3 and/or KCa3.1 channels in mice significantly decreases endothelium-dependent vasodilation and produces a hypertensive phenotype [18,19], whereas pharmacological activators of KCa channels evoke robust vasodilation in situ and acute hypotension in vivo [20,21,22,23]. Such observations demonstrate that manipulation of endothelial KCa channel activity can have major effects on the vasculature that could be exploited to mitigate endothelial dysfunction and hypertension-associated cardiovascular disease. In the present study, we have examined the effect of the KCa channel activator SKA-31 [21] on endothelium-dependent vasodilation in small resistance arteries and mean arterial blood pressure in spontaneously hypertensive rats (SHRs), a model of genetic hypertension exhibiting endothelial dysfunction [24,25,26]. Based on our results, we suggest that the SHR model exhibits altered responsiveness to the known vasodilatory actions of SKA-31.

## 2. Results

### 2.1. Functional Enhancement of Evoked Vasodilation by Acute SKA-31 Treatment

Positive modulators/activators of KCa2.x and KCa3.1 channels, such as SKA-31 [21] evoke robust vasodilation in intact arteries and organs, and lower systemic blood pressure in vivo [20,21,22,23,27,28]. In the present study, we have examined the effects of acute SKA-31 treatment on cardiovascular function in the SHR, an established model of primary hypertension that exhibits cardiac hypertrophy, heart failure, renal dysfunction, and impaired regulation of sympathetic outflow [29,30]. To examine vascular responsiveness, we performed arterial pressure myography with third order mesenteric arteries pre-constricted with phenylephrine (PE, 1 μM), and myogenically active cremaster skeletal muscle resistance arteries isolated from male SHRs (16–18 weeks of age) and aged-matched male Wistar rats. At an intraluminal pressure of 70 mmHg, the PE-induced vasoconstriction (i.e., % change of maximal passive diameter) of mesenteric arteries was not different in SHR (37.7 ± 2.5%, mean ± S.D., *n* = 3) compared with Wistar animals (32.3 ± 5.0%, *n* = 3). In cannulated cremaster resistance arteries pressurized to 70 mmHg, the developed myogenic tone in SHR vessels (50.3 ± 10.1%, *n* = 3) was similar to that observed in Wistar cremaster arteries (53.6 ± 4.8%, *n* = 4).

Stimulation of pre-constricted mesenteric arteries from SHRs with vasodilatory agents acting primarily via the endothelium (i.e., acetylcholine (ACh), bradykinin (BK) and SKA-31) revealed weaker responses to ACh (0.3 μM and 0.5 μM), BK (0.1 μM and 0.3 μM) and SKA-31 (1 μM) compared with arteries from Wistar rats, as shown by the first half of the tracings in Figure 1A,B and the histogram in Figure 1C. Giachini and colleagues have reported a similar impairment of endothelium-dependent relaxation by ACh and the KCa channel activator NS309.

In pre-constricted mesenteric arteries from stroke-prone SHRs [31]. In contrast, relaxation of mesenteric arteries by the smooth muscle-dependent agents sodium nitroprusside (10 μM) and pinacidil (5 μM) were similar in SHRs and Wistar rats (Figure 1C).

In myogenically constricted cremaster arteries from SHRs and Wistar rats, vasodilatory responses to ACh and SKA-31 were similar. However, BK (0.1 μM) evoked a profound vasoconstriction in SHR cremaster arteries, compared with the modest dilation observed in Wistar vessels (Figure 2A–C). Whereas pinacidil induced comparable inhibition of myogenic tone in SHR and Wistar cremaster arteries, SNP was less effective in SHR vessels. Collectively, these results demonstrate that mesenteric and cremaster skeletal muscle arteries from SHRs exhibited endothelial dysfunction, based on agonist-evoked vasodilatory responses, and agree with earlier studies [13].

We and others have reported that agonist-evoked, endothelium-dependent vasodilation can be enhanced in the presence of an activator of KCa2.x and KCa3.1 channels (e.g., SKA-31, NS309) [28,32,33]. In both mesenteric and cremaster arteries from SHRs, ACh-evoked vasodilation was enhanced in the presence of a threshold concentration of SKA-31 (0.3 μM) (Figure 1C and Figure 2C, yellow bars), whereas vasoactive responses to BK were unaffected. In arteries from normotensive Wistar rats that lack endothelial dysfunction, SKA-31 treatment had minimal effects on ACh and BK-evoked vasodilation (compare red and yellow bars in Figure 1C and Figure 2C).

### 2.2. Detection of BK Receptors and KCa Channels

Western blot analyses of BK receptor expression in homogenates from third order mesenteric arteries from SHR and Wistar rats revealed immune-reactive bands of ~36 kDa for the type 1 BK receptor (B1R) and ~45 kDa for the type 2 BK receptor (B2R) that were comparable to protein bands detected in homogenates prepared from human intra-thoracic arteries and human umbilical vein endothelial cells expressing native BK receptors (Figure 3A,B). SHR mesenteric arteries exhibited reduced B1R and B2R expression compared with arteries from Wistar rats (Figure 3C). In contrast, no significant differences were observed in the expression of KCa2.3 channel (~75 kDa band) and KCa3.1 channel (~45 kDa band) proteins in mesenteric arterial homogenates from the two strains (Figure 3D–F). Mesenteric arteries from SHRs further exhibited ~10-fold less mRNA for B1R and B2R compared with Wistar arteries, whereas mRNA levels for KCa2.3 and KCa3.1 channels were decreased < 2-fold, as revealed by qPCR analyses (Figure 3G).

### 2.3. Regulation of Coronary Flow in Isolated Hearts from SHRs and Wistar Rats by Endothelium-Dependent and -Independent Vasodilators

In spontaneously beating, Langendorff-perfused hearts from SHRs and Wistar rats, baseline total coronary flow rates under constant perfusion pressure were similar (14.5 ± 1.7 mL/min, *n = 3* and 13.5 ± 1.3 mL/min, *n* = 4, respectively; mean ± S.D.). Acute bolus administration of the mixed smooth muscle/endothelium-acting vasodilator adenosine produced dose-dependent, reversible increases in coronary flow that were comparable in hearts from both Wistar rats and SHRs, as shown on the left-hand side of the representative tracings in Figure 4A,B. In contrast, bolus application of the endothelium-dependent dilator BK evoked a significantly smaller increase in coronary flow in SHR hearts compared with those from Wistar rats, consistent with the presence of endothelial dysfunction (Figure 5A). SHR hearts also exhibited reduced vasodilatory responses to the smooth muscle relaxing agents pinacidil and sodium nitroprusside compared with control hearts (Figure 5A), suggesting that some coronary artery smooth muscle dysfunction also exists in SHRs. Comparable increases in coronary flow were observed in hearts from both strains in response to SKA-31, an endothelial KCa channel activator. Associated with drug-evoked elevations in total coronary flow were modest increases in left ventricular (LV) developed pressure and heart rate in both SHR and Wistar rat hearts that developed following the initial rise in total coronary flow and readily reversed as coronary flow returned to baseline. The observed increases in LV developed pressure are similar to the “Gregg Effect”, a phenomenon in which stimulated increases in coronary filling or perfusion pressure are associated with augmented contractility [35,36]. Drug-evoked changes in total coronary flow, LV developed pressure and heart rate are quantified in Figure 5A–C, respectively, and are similar to previous observations from our lab [28]. Bolus injections of saline or drug vehicle (i.e., equivalent volumes) had no effects on total coronary flow, LV developed pressure or heart rate in either strain (Figure 5A–C).

We have previously reported that addition of a threshold concentration of SKA-31 (0.3 μM) to the coronary perfusate enhanced primary vasodilatory responses to BK in Langendorff-perfused hearts from Wistar and type 2 diabetic Goto-Kakizaki rats, suggesting that SKA-31 treatment could boost endothelial function, even under conditions of type 2 diabetes [28]. In the present study, treatment of Wistar rat hearts with 0.3 μM SKA-31 had no effect on baseline coronary flow, and significantly elevated the vasodilatory response to BK, but not to the smooth muscle vasorelaxants adenosine, pinacidil and SNP (Figure 5A). Associated changes in LV developed pressure and heart were also modestly enhanced with SKA-31 treatment. In SHR hearts, however, administration of SKA-31 did not modify changes in total coronary flow evoked by BK, adenosine, pinacidil or SNP, even though bolus administrations of SKA-31 itself evoked similar increases in total coronary flow in hearts from both strains.

### 2.4. Effect of Acute SKA-31 Administration on Systemic Blood Pressure in SHRs

The ability of SKA-31 to directly dilate isolated arteries and the coronary circulation of SHRs (Figure 1, Figure 2 and Figure 4) suggested that it may also lower systemic blood pressure in vivo. In conscious SHRs instrumented with radio-telemeters (*n = 3*), we observed mean arterial pressure (MAP) and heart rate (HR) values of ~130 mmHg and 290 beats per minute (bpm) over a 4 h baseline period. These values are in line with those previously measured in similarly aged male SHRs by radio-telemetry [37]. Acute intraperitoneal (i.p.) injection of drug vehicle alone produced an immediate and transient spike in MAP and HR that reversed within 1 h (Figure 6). Following 2–3 days of rest, administration of SKA-31 (30 mg/kg) to the same SHRs elicited similar spikes in MAP and HR. However, these increases did not readily reverse and were sustained above baseline over the four-hour period following injection. The eight-hour window selected for MAP and HR measurements occurred entirely within the daily light cycle and was chosen to minimize possible complications associated with diurnal rhythms and light/dark activity transitions. Average values of MAP and HR over this four hour window were significantly elevated in SHRs following SKA-31 injection (i.e., 144 ± 12 mmHg and 319 ± 14 bpm, respectively; mean ± S.D.) compared with values recorded from the same animals in response to vehicle injection (MAP, 130 ± 9 mmHg; HR, 287 ± 10 bpm) (Figure 6).

## 3. Discussion

Systemic hypertension is major risk factor in the development of cardiovascular disease and is further associated with endothelial dysfunction, which compromises NO bioavailability and the endothelial regulation of vascular tone [4,5,6]. Small molecule activators of KCa2.x and KCa3.1 channels (e.g., SKA-31, NS309) are known to evoke vasodilation in healthy [27,38] and diseased arteries/vascular beds [28,33], and reduce systemic blood pressure in both small and large mammals [20,21,22,23]. In the present study, we hypothesized that SKA-31 treatment may have positive actions on vascular function and mean arterial pressure in adult SHRs, a well-characterized animal model of primary hypertension that also exhibits endothelial dysfunction [29,39,40]. Our results describe the vascular responsiveness to SKA-31 treatment in situ, along with its effects on systemic blood pressure and heart rate in 16–18 week old, male SHRs.

The main findings of our study are as follows: (1) Vasodilatory responses to endothelium- (i.e., ACh, BK) and smooth muscle-dependent agents (i.e., pinacidil and SNP) were either decreased or absent in small arteries from the mesentery and cremaster skeletal muscle, and the coronary circulation of SHRs, compared with Wistar control rats (Figure 1, Figure 2, and Figure 4). In contrast, bath addition of the KCa channel activator SKA-31 evoked either similar or modestly reduced vasodilatory responses in equivalent vascular tissues from both strains. (2) The expression of type 1 and type 2, G-protein-coupled BK receptors (B1R and B2R, respectively) was decreased at both the protein and mRNA levels in small mesenteric arteries from SHRs compared with Wistar controls, whereas KCa2.3 and KCa3.1 channel expression was comparable (Figure 3). (3) Langendorff-perfused, spontaneously beating hearts from SHRs exhibited reduced vasodilatory responses to BK, SNP, and pinacidil. Treatment with a threshold concentration of SKA-31 (0.3 μM) failed to augment BK-evoked increases in coronary flow, whereas this same intervention enhanced BK-stimulated coronary flow in Wistar control hearts. SKA-31 treatment also augmented ACh-mediated dilation in mesenteric arteries from Wistar rats and SHRs (Figure 1C). 4) Acute bolus administration of SKA-31 (30 mg/kg, I.P injection) to SHRs elevated mean arterial blood pressure, whereas this same dose of SKA-31 is reported to evoke acute hypotension in other rodent models of hypertension [21,22].

Our data demonstrating reduced endothelium-dependent vasodilation in mesenteric and cremaster arteries, and the coronary circulation of SHRs are consistent with earlier reports describing impaired EDH [41] and vascular reactivity in these animals [31,42,43,44,45]. The widespread nature of this impairment in SHR vasculature is reminiscent of observations in patients showing that coronary endothelial dysfunction co-associated with similar dysfunction in brachial arteries [46]. Our results further demonstrate impaired vascular smooth muscle function in the coronary circulation of SHR hearts, based on decreased vasodilatory responses to the direct smooth muscle relaxants SNP and pinacidil (Figure 4 and Figure 5). Remarkably, BK-evoked vasodilation was decreased significantly in all three vascular preparations from SHRs, which was paralleled by reduced expression of BK receptors in small mesenteric arteries. We speculate that a similar reduction in the level of endothelial BK receptors (i.e., B2 subtype) may account for the weak vasodilation noted in the coronary circulation, along with the overt vasoconstriction observed in cremaster resistance arteries in response to BK stimulation. Functionally, B2 receptors present on the vascular endothelium evoke robust vasodilation, whereas smooth muscle B2 receptors promote contraction [47,48,49,50]. In support, Nawa and colleagues have shown that BK-induced vasodilation and vasoconstriction in the perfused rat mesenteric bed occurs via B2 receptors located on the vascular endothelium and smooth muscle, respectively [51]. Thus, impaired B2 receptor signaling in the endothelium of SHR arteries could unmask vasoconstrictor responses to BK via direct actions on smooth muscle [25]. In the absence of overt inflammation, B1 receptors are reported to play little to no role in BK-stimulated vasoactive responses [48,49,50,52]. It is noteworthy that ACh-evoked vasodilation was reduced in small mesenteric arteries from SHRs, compared with Wistar rats, yet was largely preserved in cremaster arteries. Such observations suggest that GPCR signaling pathways in the endothelium of discrete vascular beds are impacted differentially by SHR-associated vascular pathology.

In contrast to the observed heterogeneous impairment of endothelium-dependent, GPCR-stimulated vasodilation, the KCa channel activator SKA-31 evoked similar or slightly reduced vasodilatory responses in mesenteric, cremaster and coronary artery preparations from SHRs compared with Wistar rats (Figure 1, Figure 2, and Figure 4). These findings suggest that endothelial KCa channel activity and functional vasoactive coupling of these channels to surrounding smooth muscle [12,38] are similar in diverse vascular beds between the normo- and hypertensive rats. This interpretation is supported by the comparable expression of KCa2.3 and KCa3.1 channel protein and mRNA observed in small mesenteric arteries from Wistar rats and SHRs (Figure 3). In contrast, Weston and co-workers have reported that KCa2.3 channel expression in SHR mesenteric arteries is decreased ~2-fold relative to Wistar Kyoto rats [41].

We and others have reported that endothelium-dependent, agonist-evoked vasodilation can be augmented in the presence of a low or threshold concentration of a KCa channel activator, such as SKA-31 [28,32,33]. This treatment strategy can further mitigate endothelial dysfunction associated with type 2 diabetes [28,33] and we speculate that it occurs via a pharmacological “priming” or sensitization of endothelial KCa channel activity and EDH-type signaling [53]. In small mesenteric arteries from both SHRs and Wistar rats, SKA-31 treatment augmented ACh, but not BK, induced vasodilation (Figure 1), consistent with the putative sensitization of this GPCR signaling pathway in these vessels. However, SKA-31 treatment was unable to enhance the BK-mediated increase in coronary flow in SHR hearts, whereas this response was significantly elevated in Wistar controls (Figure 4 and Figure 5). We speculate that an impairment of BK receptor signaling in the coronary vasculature of SHRs (e.g., decrease in receptor expression, impaired downstream cascade, etc.) may preclude any augmentation of this pathway by SKA-31 treatment, even though endothelial KCa channel activity itself remained intact. It is further possible that endothelial KCa channels in the coronary vasculature of SHRs may be mis-localized in the plasma membrane, and unable to respond properly to BK-evoked elevations in cytosolic calcium. Such a scenario could preclude a positive effect imparted by SKA-31 induced sensitization.

Earlier studies have reported that acute in vivo administration of SKA-31 lowers mean arterial pressure (MAP) in both normo- and hypertensive animal models [20,21,22,23]. It was thus surprising that a comparable dose of SKA-31 (i.e., 30 mg/kg) did not evoke a pronounced hypotensive response in SHRs following bolus i.p. injection (Figure 6), particularly in view of its vasorelaxant effects in isolated vascular preparations. One possibility is that SKA-31 administration produced a modest reduction in peripheral vascular resistance and MAP, triggering a compensatory increase in HR and cardiac output that overcompensated for the initial hypotensive action of SKA-31 (Figure 6A provides a hint that SKA-31 decreased MAP within the first 30 min following injection). We speculate that dysfunction within the vasculature, the heart or autonomic regulation of these systems in the SHR may contribute to such homeostatic overcompensation, and lead to an exaggeration of the MAP response in magnitude and/or duration. Another possibility is that SKA-31 administration led to a desensitization of endothelial KCa channel activities, leading to a reduction of vasodilatory capacity and elevation of MAP. Based on these observations, we suggest that different rodent models of hypertension (e.g., spontaneous vs. genetically engineered vs. pharmacologically induced) exhibit distinct hemodynamic responses to SKA-31 administration in vivo. As the hemodynamic actions of acutely administered SKA-31 may be affected by multiple factors (e.g., hormonal and/or autonomic influences, peripheral vs central sites of actions, pharmacokinetics), some of these factors may differ in the SHR and contribute to our observations.

In summary, SHRs exhibited differing degrees of vascular endothelial dysfunction, the nature of which depended upon the particular arterial bed (i.e., mesentery vs. cremaster skeletal muscle vs, coronary). Whereas these isolated preparations demonstrated comparable vasodilatory responses to the KCa channel activator SKA-31 in situ, acute i.p. administration of SKA-31 in vivo did not lower MAP in an obvious manner, as has been reported in other rodent models of hypertension. Further investigation is required to elucidate the molecular defect(s) underlying impaired BK-mediated vasodilation and how the vascular and systemic actions of KCa channel activators in vivo are influenced by the underlying cause and extent of primary hypertension.

## 4. Materials and Methods: 

### 4.1. Animals and Treatment

All procedures involving animal usage and treatment were approved by University of Calgary Animal Care Committee (project code: AC16–0170, approved 2 November, 2016, Health Sciences Animal Care Committee) and conducted in accordance with the current guidelines of the University of Calgary and the Canadian Council for Animal Care. Two groups of male animals, Wistar rats (16–17 weeks of age) and SHRs (16–18 weeks), from Charles River Laboratories (Montréal, PQ were used for the study. Rats were housed in single cages, 12 h light/dark cycle with water and chow freely available. Rats were treated with either drug vehicle or 30 mg/kg SKA-31 by acute i.p. injection. SKA-31 (naphtho[1,2-d]thiazol-2-ylamine) was synthesized as previously described [21]. Euthanasia was performed by a bolus i.p. injection of sodium pentobarbital (50–60 mg/kg); once Stage 3 anesthesia was achieved; the chest was opened surgically and the heart rapidly removed. All additional tissues and organs were collected according to the experimental protocols.

### 4.2. Surgical Procedure for Telemeter Implantation in SHRs

Analgesia was achieved by the administration of the analgesics buprenorphine 0.05 mg/kg (subcutaneous) both pre-and post-surgery, and a single dose of ketamine/xylazine (100 mg/kg, intramuscular) administered pre-operatively. SHRs were then anesthetized using inhalational isoflurane. Following anesthesia, hydration was maintained by administering sterile saline to animals. Briefly, rats were placed on a heating pad (37 °C) and positioned dorsally throughout the surgical period. Laparotomy was performed in a sterile surgical environment and the aorta and vena cava were exposed. The section of aorta distal to the renal bifurcation and proximal to the iliac bifurcation was gently separated from surrounding tissue and vena cava. Sutures were used to retract the aorta, and a 21 g needle was used to gently pierce the vessel. A PAC-40 telemeter (DSI International, Minneapolis, MN, USA) was inserted and fixed in place with Vetbond, as per manufacturer’s instructions, before implantation of the radio-telemeter. A piece of micro-cellulose paper was then placed over top of the catheter to promote tissue growth. The body of the telemeter was stitched into the gut wall and the opening was then closed in layers [37,54,55]. Buprenorphine (0.05 mg/kg, subcutaneous) was then administered for 3 days (twice daily) and the rats were allowed 14 days of recovery prior to basal blood pressure recording. Animals were monitored daily by either veterinarians or trained technicians throughout the study for clinical signs of disease, such as weight loss, intractable diarrhea and opportunistic infection.

### 4.3. Radio-Telemetry and Data Acquisition

Blood pressure data (mmHg) were recorded every 5 min (5 s sampling period) over the experimental period using computer-based acquisition hardware and software (Ponemah v6.2 Telemetry System, Data Sciences International, Minneapolis, MN, USA). To evaluate the effect of a bolus i.p. injection of SKA-31 on blood pressure, SHRs were treated with either drug vehicle or 30 mg/kg SKA-31 [21]. Injections of either vehicle or SKA-31 were typically performed during the mid-point of the 12-h light period, and the displayed data in Figure 6 were all collected during light conditions to avoid potential effects of diurnal behavior on cardiovascular function. Note that for each instrumented SHR, the initial administration of drug vehicle and SKA-31 was followed by a second round of injections approximately one week later, in which the order of compound administration was reversed. The data collected during these two injection cycles for each animal were then averaged as technical replicates.

### 4.4. Protocol for Langendorff-Perfused, Isolated Heart Experiments

Prior to euthanasia, SHRs and Wistar rats were injected i.p. with heparin (50 IU) to minimize blood clotting in the coronary vasculature. Animals were then injected with sodium pentobarbital (50 mg/kg) to induce Stage 3 anesthesia, then sacrificed by rapid removal of the heart. Following excision and cleaning, isolated hearts were mounted on a blunted 16-gauge needle and perfused retrograde through the ascending aorta under constant pressure (gradient ~90 cm H_2_O) with oxygenated (95% O_2_/5% CO_2_) Krebs’ buffer maintained at 37 °C by a heated water jacket system, as described [56]. Coronary flow rate was monitored via an inline Doppler flow probe and a Transonic T206 flow meter. Iso-volumetric left ventricular (LV) developed pressure was measured with a fluid-filled latex balloon inserted into the left ventricle via the left atrium and connected to a pressure transducer (model 60–3002, Harvard Apparatus, Holliston, MA, USA). The balloon volume was adjusted to obtain an end diastolic pressure of 15–20 mmHg. Coronary flow and LV developed pressure data were digitally recorded at a sampling frequency of 10 Hz using a PowerLab 4/26 DAQ acquisition system (ADInstruments, Colorado Springs, CO, USA). Langendorff-perfused hearts were allowed to equilibrate until heart rate and LV contractility reached a steady-state level (i.e., 20–30 min). Experimentally, vehicle and drugs were acutely administered as 0.1 mL bolus injections into the aortic perfusate via an injection port positioned immediately upstream of the heart.

### 4.5. Arterial Pressure Myography

The mesentery and cremaster skeletal muscle were excised and placed in a cooled dissection chamber containing ice-cold Krebs’ solution (in mM): 115 NaCl, 4.7 KCl, 1 MgSO_4_, 1 NaH_2_PO_4_, 25 NaHCO_3_, and 2 CaCl_2_, pH adjusted to 7.4 with 1M NaOH. The third to fourth order branches of the mesenteric arterial tree (300–350 micron internal passive diameter) were dissected, cleaned of adherent connective tissue, and transferred to an arterial pressure myography chamber (Living Systems, Burlington, VT, USA). Individual vessels (~2 mm in length) were mounted on two glass cannulae fitted in the myography chamber, with each vessel end secured on a cannula by a 10.0 suture thread. The vessel lumen was filled with Krebs’ solution containing 1% *w*/*v* bovine serum albumin. The cannulated artery was placed on the stage of an inverted microscope and superfused with Krebs’ buffer gassed with 95% air/5% CO_2_ at a flow of ~7 mL/min using a peristaltic pump. A glass heat exchanger was used to warm the bubbled Krebs’ solution to a bath temperature of 36–37 °C. After ~10 min equilibration, the intraluminal pressure of cannulated vessels was increased and then maintained at 70 mmHg throughout the experiment (no intraluminal flow was used). Cremaster arteries (branch 1A or 2, ~200 micron maximal passive intraluminal diameter) were isolated and mounted in a similar manner. Following 15–20 min at 70 mmHg intraluminal pressure, cremaster arteries typically developed myogenic tone (i.e., 30%–50% of maximal passive diameter). Mesenteric arteries developed only a modest degree of pressure-induced tone (typically < 15%) and were then stimulated with 1μM PE to increase the extent of contraction. Continuous video measurement of the intraluminal vessel diameter was performed using a diameter tracking system (IonOptix, Milton, MA, USA). Pharmacological agents were added to the bath perfusate, as described [27,38].

### 4.6. Quantitative PCR

Third and fourth order mesenteric arteries were dissected from SHRs and Wistar rats, followed by removal of surrounding connective tissues. Total RNA was then prepared from a total weight of ~0.5 mg of vessels from each animal (*n* = 4–5 animals per group), using a RNeasy micro kit (Qiagen) according to the manufacturer’s instructions. A cDNA synthesis kit (Quantinova, Qiagen, Montreal, PQ, Canada) was used to prepare cDNA from total RNA. Real-time PCR was performed using the Kapa SYBR Fast Universal qPCR kit (Kapa Biosystems, Wilmington, MA, USA) and validated primers from Integrated DNA Technologies. Rat glyceraldehyde 3-phosphate dehydrogenase (GAPDH) gene was used as the internal reference gene and counter checked with β-actin (Actb). Control reactions and those containing cDNA from mesenteric arteries were performed with 1 ng of template per reaction. The running protocol was performed using an Eppendorf Realplex 4 Mastercycler instrument and extended to 45 cycles, with each cycle consisting of the following heating steps: 95 °C for 5 s, 55 °C for 10 s, and 72 °C for 8 s. PCR specificity was checked by dissociation curve analysis. Assay validation was confirmed by testing serial dilutions of pooled template cDNAs, indicating a linear dynamic range over 2.8–0.0028 ng of template and yielding percent efficiencies ranging from 100%–118%. Control samples lacking template yielded no detectable fluorescence. Expression levels of mRNA were determined and quantified using the 2^−ΔΔCT^ method and the relative expression software tool (REST) version 2.0.13 [34]. Note that reporting of qPCR data has been performed in accordance with the MIQE guidelines [57].

### 4.7. Preparation of Vessel Homogenates

Four to five mesenteric arteries rings of similar size were washed with distilled water and promptly transferred to a buffer containing: 50 mM Tris HCl (pH 7.4), 150 mM NaCl, 1% (*v*/*v*) Triton-X-100, 1 mM EGTA (ethylene glycol-bis(2-aminoethylether)-N,N,N’,N’-tetraacetic acid), 1 mM EDTA (ethylenediaminetetraacetic acid), 1 mM PMSF (phenylmethanesulfonyl fluoride), 5 μg/mL leupeptin, 5 μg/mL pepstatin A, 5 μg/mL aprotinin, and 100 mM dithiothreitol. Arterial segments were disrupted using a tight-fitting Teflon homogenizer in a glass vessel and placed on ice for 30 min. After centrifugation at ~6000×g for 5 min at 4 °C, supernatants containing the microsomal and cytosolic fractions were aliquoted into small Eppendorf tubes and snap-frozen at −80 °C. cDNA clones encoding human KCa2.3 and KCa3.1 channels were transiently expressed in HEK 293 cells using Lipofectamine 2000. After ~48 h in culture, transfected cells were harvested and soluble lysates were prepared, as described [58]. HEK 293 cells transfected with cDNA encoding the pore-forming subunit of the murine Kir2.1 channel were used as a negative control. Homogenates prepared from human umbilical vein endothelial cells (EA.hy926 cell line) [59] and human internal thoracic arteries (ITAs) obtained from consented patients undergoing coronary artery bypass surgery were used as positive controls for BK receptors B1R and B2R. All subjects gave their informed consent for inclusion before they participated in the study. This work was conducted in accordance with the Declaration of Helsinki, and the protocol for tissue collection was approved by the Conjoint Health Research Ethics Board, University of Calgary, protocol REB15–1364.

### 4.8. SDS-PAGE and Western Blot Analyses

Protein homogenates prepared from mesenteric resistance arteries, cultured cells and human ITAs were separated by SDS polyacrylamide gel electrophoresis using a 10% separating gel. Resolved proteins were electro-transferred to nitrocellulose membrane (0.2 μm pore size) for 2 h at 70 V and 4 °C using a Hoefer TE22 transfer tank. Transfer buffer contained 25 mM Tris, 192 mM glycine, 0.1% SDS (*w*/*v*) and 20% (*v*/*v*) methanol. Membranes were blocked for 2 h at room temperature in Tris-buffered saline + 0.1% Tween-20 (*v*/*v*) (TBS-T, pH 7.4) containing 5% (*w*/*v*) skim milk powder, and then incubated overnight at 4 °C with a primary antibody against one of the following: KCa2.3 channel (Alomone Labs, Jerusalem, Israel, catalogue #APC-025, 1:500 dilution), KCa3.1channel (Santa Cruz Biotechnology, Santa Cruz, CA, USA, catalogue #sc-365265, 1:1000 dilution), bradykinin B1R (Santa Cruz Biotechnology, catalogue #sc-293196, 1:1000), bradykinin B2R (Santa Cruz Biotechnology, catalogue #sc-136216, 1:1000) or β-actin (Santa Cruz Biotechnology, Santa Cruz, CA, USA, catalogue #sc-47778, 1:10,000) diluted in TBS-T + 1% (*w*/*v*) skim milk powder. Membranes were washed with TBS-T for 3 × 5 min at room temperature on a platform rocker, and then incubated for 1–2 h at room temperature in TBS-T + 1% (*w*/*v*) skim milk powder containing peroxidase-conjugated goat anti-mouse or anti-rabbit IgG (Millipore Sigma, St. Louis, MO, USA, catalogue #s AP124P and AP132P, 1:5000 dilutions).

## Figures and Tables

**Figure 1 ijms-20-03481-f001:**
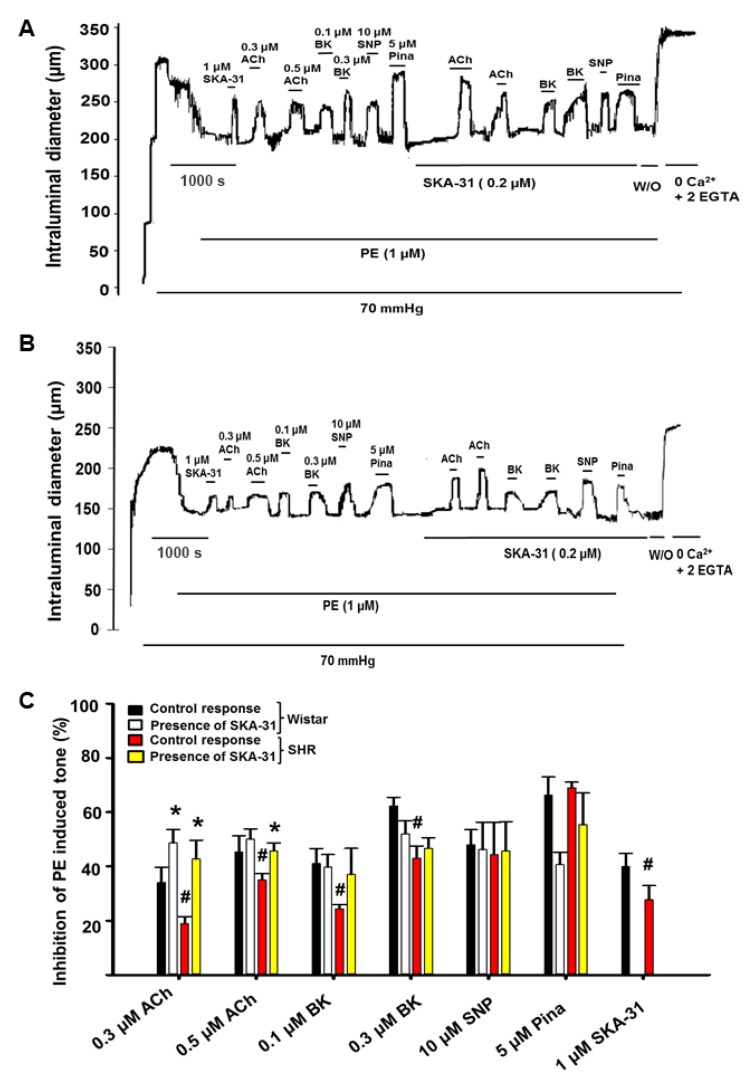
Treatment of small mesenteric arteries from Wistar (**A**) and spontaneously hypertensive rats (SHRs) (**B**) with SKA-31 improves agonist-evoked, endothelium-dependent vasodilation (**C**). Panels A and B display representative tracings of vasorelaxation evoked by the KCa channel activator SKA-31 (1 μM), acetylcholine (ACh, 0.3 and 0.5 μM), bradykinin (BK, 0.1 and 0.3 μM), sodium nitroprusside (SNP, 10 μM), and pinacidil (Pina, 5 μM) in cannulated and pressurized mesenteric arteries from a Wistar rat (**A**) and SHR (**B**) that have been pre-constricted with PE. The horizontal bars above the intraluminal diameter tracing indicate bath application of individual agents. In the second half of the protocol, arteries were superfused with 0.2 μM SKA-31 (denoted by horizontal bar below the tracing) to evaluate the potential augmentation of evoked vasorelaxation. The histogram in panel (**C**) quantifies the average inhibition of PE induced tone in isolated mesenteric arteries from Wistar rats (black bars) and SHRs (red bars) in either the absence or presence of 0.2 μM SKA-31 (white and yellow bars, respectively). Data are plotted as mean ± S.D., with *n* = 3 animals for each experimental condition. The hashtag symbol (#) denotes a statistically significant difference in the magnitude of inhibition compared with the control condition (Wistar), as determined by one-way ANOVA and a Tukey’s post-hoc test, *p* < 0.05. The asterisk (*) denotes a statistically significant difference vs. the response observed in the absence of SKA-31 treatment, as determined by a paired Student’s t-test, *p* < 0.05.

**Figure 2 ijms-20-03481-f002:**
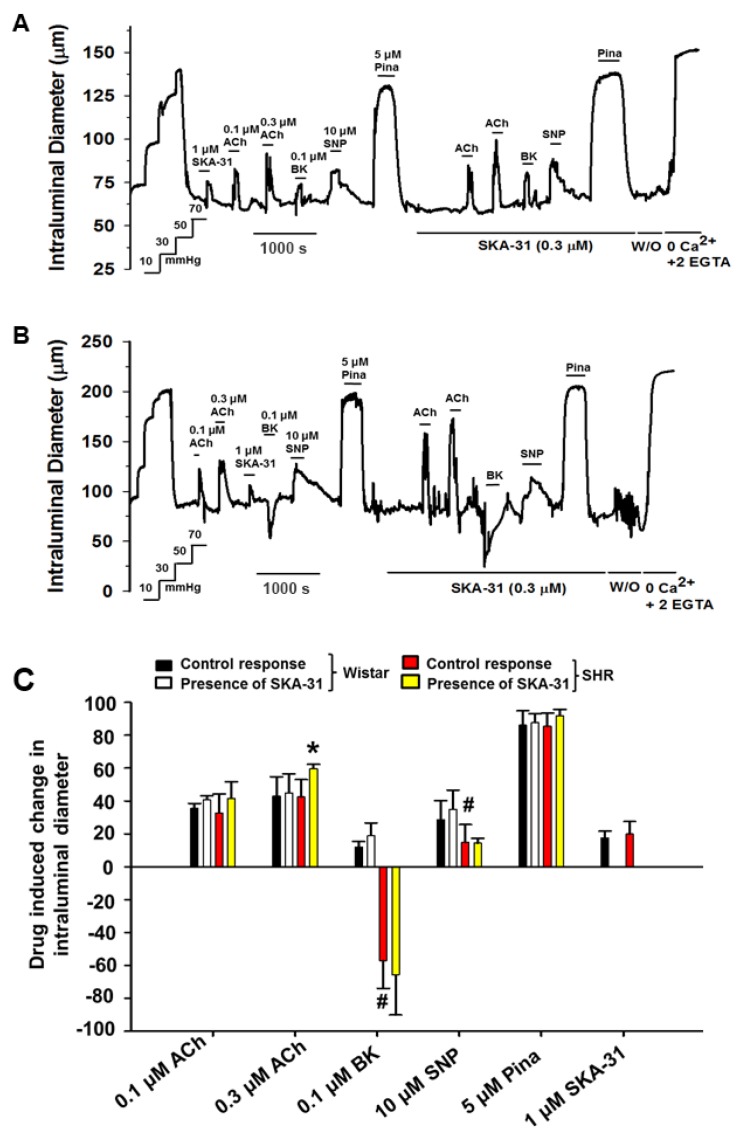
Treatment of cremaster skeletal muscle arteries from Wistar rats (**A**) and SHRs (**B**) with SKA-31 has minimal effect on endothelium-dependent vasodilation. The representative tracing in panel A depicts vasodilatory responses evoked by bath application of acetylcholine (ACh, 0.1 and 0.3 μM), bradykinin (BK, 0.1 μM), sodium nitroprusside (SNP, 10 μM) and pinacidil (Pina, 5 μM) in a myogenically constricted cremaster artery from a Wistar rat, in the absence and presence of 0.3 μM SKA-31, as indicated by the horizontal bars above and below the tracing. Panel B displays vasoactive responses to the same agents in a myogenically constricted cremaster artery from a SHR. The summary data in panel (**C)** display the percent change in intraluminal diameter evoked by ACh, BK, SNP, and Pina in the absence and presence of SKA-31. Data are plotted as means ± S.D., with *n* = 3–4 animals for each experimental condition. The hashtag symbol (#) indicates a statistically significant difference in the magnitude of the vasoactive response compared with the control condition (Wistar), as determined by one-way ANOVA and a Tukey’s post-hoc test, *p* < 0.05. The asterisk (*) denotes a statistically significant difference vs. responses in the absence of SKA-31 exposure, as determined by a paired Student’s t-test, *p* < 0.05.

**Figure 3 ijms-20-03481-f003:**
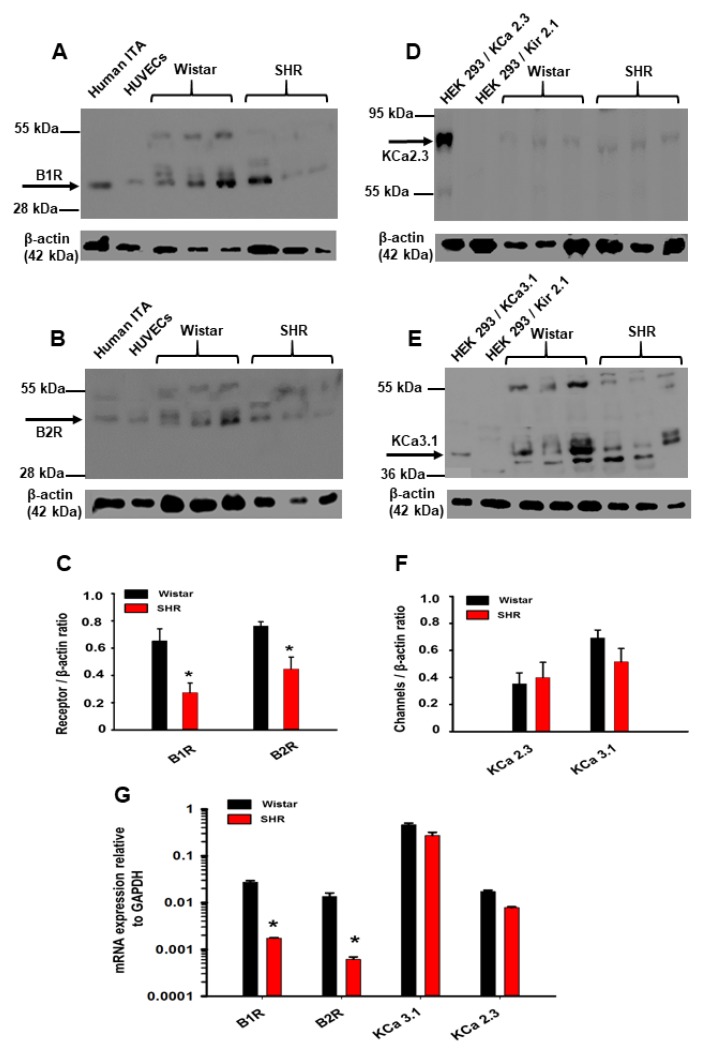
Molecular detection of bradykinin (BK) receptors and KCa channels in Wistar and SHR mesenteric arteries. The protein expression of BK receptor type 1 (B1R, panel (**A**)) and type 2 (B2R, panel (**B)**), along with KCa2.3 and KCa3.1 channels (panels (**D)** and (**E)**) was analyzed by western blot in mesenteric artery homogenates derived from age-matched SHRs and Wistar rats. Homogenates prepared from human intra-thoracic arteries (ITA) and human umbilical vein endothelial cells (HUVECs) served as positive controls for the detection of B1R and B2R proteins. Lysates prepared from HEK 293 cells transfected with cDNA encoding either KCa3.1 or KCa 2.3 channels were used as positive controls and are displayed in lane 1 of the corresponding blots. HEK 293 cells transfected with cDNA encoding mouse Kir2.1 channel served as a negative control. Quantification of protein expression for B1R and B2R, and KCa2.3 and KCa3.1 channels is shown in panels (**C)** and (**F)**, respectively. Staining intensities of selected immuno-reactive bands are expressed as a ratio with detected β-actin expression in the same homogenate. The asterisk (*) signifies a statistical difference between the indicated groups (*n* = 4–5 animals), as determined by a Student’s t-test, *p* < 0.05. Panel (**G)** quantifies the levels of mRNA encoding B1R, B2R, KCa2.3 channel, and KCa3.1 channel in mesenteric arteries from SHRs and Wistar rats, as determined by real-time, quantitative PCR analyses. The ratio of mRNA expression for each target in the Wistar and SHR mesenteric arteries was calculated using REST software [34]. Detection of GAPDH mRNA was utilized as an internal reference in all assays. Data are expressed as means ± S.D. and statistical analyses were determined by one-way ANOVA and a Tukey’s post-hoc test (*n* = 4–5 animals per group).

**Figure 4 ijms-20-03481-f004:**
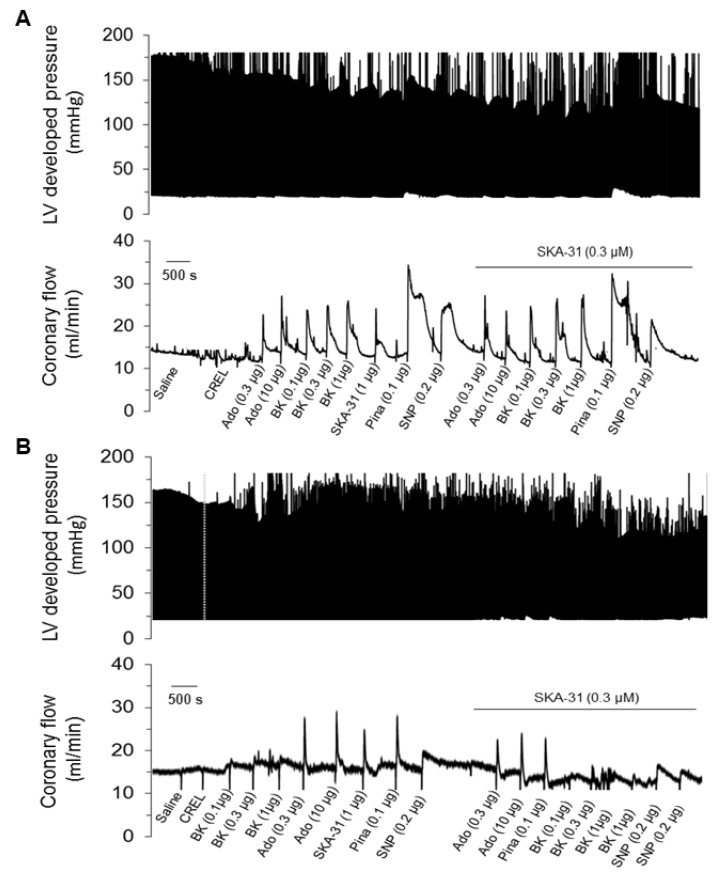
Stimulus-evoked increases in coronary flow are blunted in Langendorff-perfused hearts from SHRs. Panel (**A**) displays representative tracings of LV developed pressure (upper) and coronary flow (lower) in a spontaneously beating, Langendorff-perfused male Wistar rat heart in response to single bolus administrations of saline, drug vehicle (CREL), bradykinin (BK), adenosine (ADO), SKA-31, pinacidil (Pina), sodium nitroprusside (SNP), as denoted below the flow tracing. The repeated administration of these compounds in the presence of 0.3 μM SKA-31 is indicated by the horizontal bar above the flow tracing. Recordings of LV developed pressure and coronary flow in an isolated, perfused heart from an age-matched male SHR are shown in panel (**B**). Data are representative of 3–4 isolated heart preparations from each strain subjected to the same experimental protocol.

**Figure 5 ijms-20-03481-f005:**
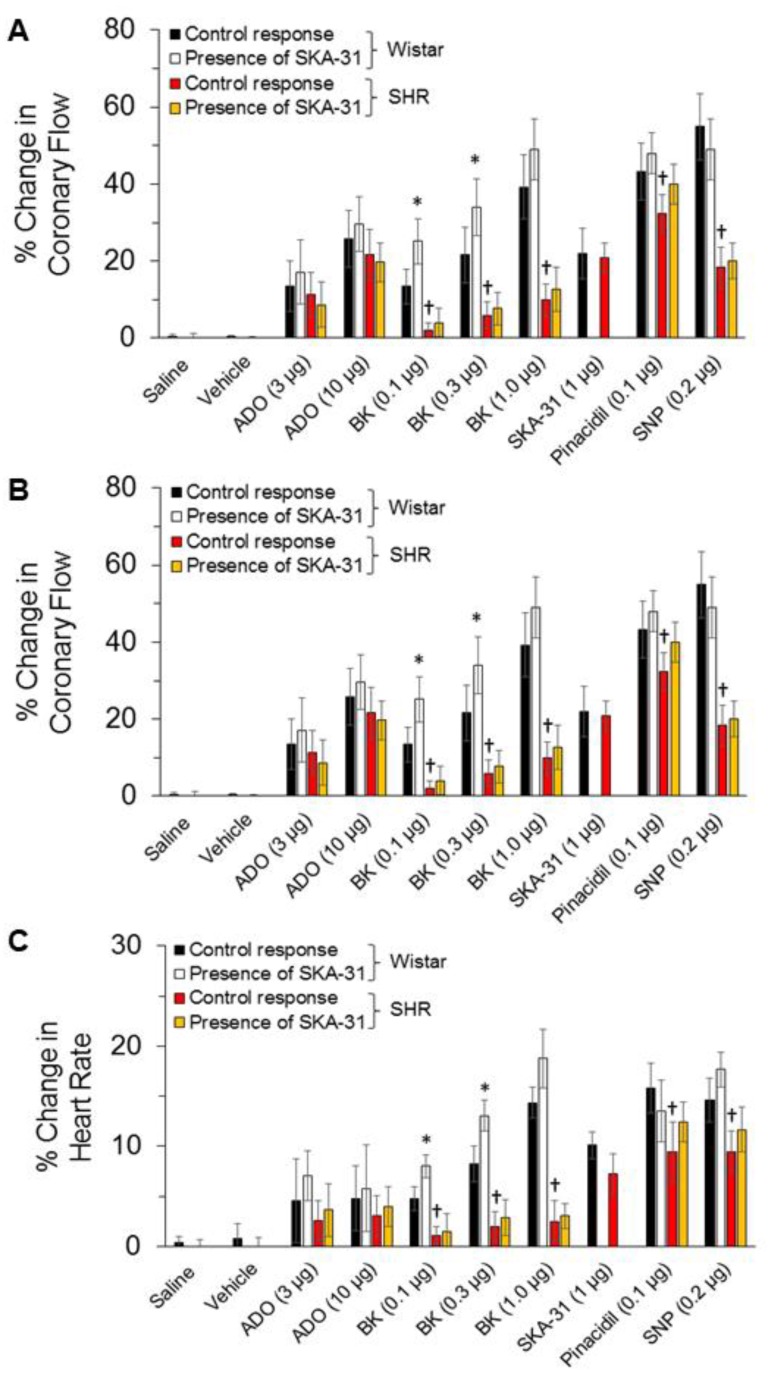
Quantification of drug-induced changes in coronary flow, left ventricular developed pressure, and heart rate in SHR and Wistar rat hearts. Histograms quantify the stimulus-induced changes in total coronary flow (panel (**A**), left ventricular (LV) developed pressure (**B**) and heart rate (**C**) following acute bolus administration of bradykinin (BK, 0.1, 0.3, and 1 µg), adenosine (ADO, 0.3 and 10 µg), SKA-31 (1 µg), pinacidil (Pina, 0.1 µg), and SNP (0.2 µg) in spontaneously beating, Langendorff-perfused male SHR and age-matched Wistar rat hearts. Values are expressed as a percentage increase in the indicated parameter relative to the baseline value recorded immediately prior to the administration of a given compound. Data are presented as mean ± SD and were analyzed by one-way ANOVA, followed by a Tukey’s post-hoc test. An asterisk (*) denotes a significant difference in the magnitude of evoked response to an individual drug for a given strain; the † symbol indicates a statistically significant difference in the magnitude of responses between the Wistar and SHR hearts; *p* < 0.05.

**Figure 6 ijms-20-03481-f006:**
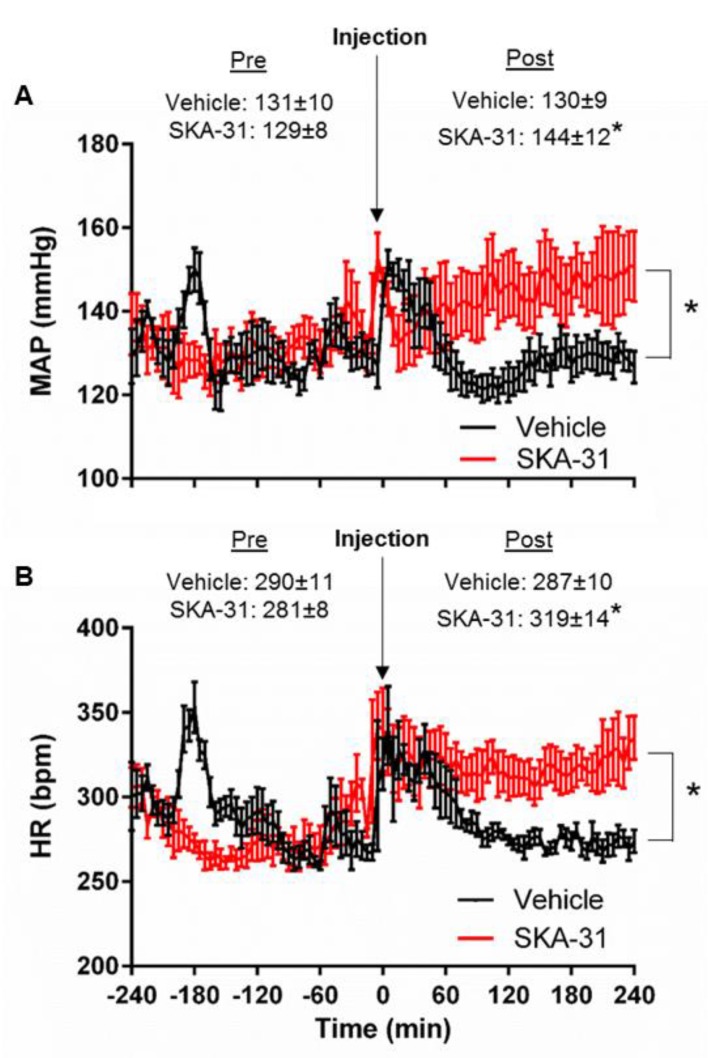
Changes in mean arterial pressure and heart rate following acute administration of SKA-31 to SHRs. Mean arterial blood pressure (MAP) and heart rate (HR) were continuously monitored in SHRs surgically implanted with radio-telemeters prior to and following administration of either drug vehicle or SKA-31 (30 mg/kg). Panels (**A**) and (**B**) plot average values for MAP (in mmHg) and HR (in beats per minute), respectively, over a four-hour period before and after an acute i.p. injection of either drug vehicle or SKA-31; the vertical arrow indicates the time of injection. Data for vehicle injections are denoted by black symbols; the red symbols indicate SKA-31 administration. Each animal was injected with drug vehicle and SKA-31, and the two injections were separated by 2–3 days. This treatment cycle was repeated twice for each animal over a two-week period. Data are plotted as mean ± SD at each time point. The asterisk (*) indicates a statistically significant difference between the vehicle and SKA-31 treatment post-injection, as determined by a two-way, repeated measures ANOVA, *p* < 0.05. No statistical difference is observed in the mean data for either MAP or HR obtained pre-injection.

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
