# Peer review of "Pharmacological Targeting of KCa Channels to Improve Endothelial Function in the Spontaneously Hypertensive Rat"

_ijms, 2019, doi:10.3390/ijms20143481_

Round 1

Reviewer 1 Report

This paper examines aspects of the different degrees of vascular endothelial dysfunction across various tissue beds in SHR and finds that despite comparable vasodilator responses to calcium-activated potassium channel agonsit SKA-31 in vitro, in vivo administration did not lower mean arterial pressure in SHRs. This differs from other rat models of hypertension. Thus, SHRs appear less responsive to endothelial calcium-activated potassium channel activators.

Throughout, please do not keep redefining abbreviations! The authors must go through their manuscript carefully and delete for example “spontaneously hypertensive rat(s)” the 2nd, 3rd, etc time the full term appears. Only “SHR” should be shown subsequent to its first appearance after the full term. The same applied to “nitric oxide (NO)”, but there are others that need to be fixed as well.

Line

24: Authors must learn that they must NEVER use “however” as a conjunction. MUST start a NEW SENTENCE! --- “…. from SHRs. However, responses …”

(The authors should be congratulated for correctly using “however” in line 231.)

25, 342, 368: Change to “i.p.”

26: There is no such word in the English language as “pharmacologic”. This is dreadful American jargon! Change to “pharmacological”.

56-66: Delete. This is a summary of the results and must not appear in the Introduction.

239: Delete “observed” as it is redundant.

243: Once again the use of “however” is incorrect. The word ‘however” may NEVER be used to joing two sentences. Must start a new sentence!

390: “sec” is a trigonometric term and must never be used as the abbreviation for “seconds”. The SI abbreviation is “s”. Here use “seconds”. In 451 the authors use both “s” and “sec”!?!

392: Why “intraperitoneal” here but “i.p.” elsewhere?

REFERENCES

These need to be edited so that they conform with journal style. Titles of journals must be capitalized. If journal style requires abbreviated journal names. 

Article titles that are capitalized (e.g., refs 1, 2, 12, 13) must be changed to lower case.

592: Should this journal name include “Scandinavica”?

Author Response

Responses to Queries from Reviewer #1

Open Review

English language and style

( ) Extensive editing of English language and style required
( ) Moderate English changes required
(x) English language and style are fine/minor spell check required
( ) I don't feel qualified to judge about the English language and style

Yes

Can   be improved

Must   be improved

Not   applicable

Does   the introduction provide sufficient background and include all relevant   references?

(x)

( )

( )

( )

Is   the research design appropriate?

(x)

( )

( )

( )

Are   the methods adequately described?

(x)

( )

( )

( )

Are   the results clearly presented?

(x)

( )

( )

( )

Are   the conclusions supported by the results?

(x)

( )

( )

( )

 We thank the reviewer for his/her constructive comments and overall positive assessment of our submitted manuscript. Our responses to individual concerns are presented below. Note that major revisions to the text are highlighted in yellow to facilitate identification.

Comments and Suggestions for Authors

This paper examines aspects of the different degrees of vascular endothelial dysfunction across various tissue beds in SHR and finds that despite comparable vasodilator responses to calcium-activated potassium channel agonsit SKA-31 in vitro, in vivo administration did not lower mean arterial pressure in SHRs. This differs from other rat models of hypertension. Thus, SHRs appear less responsive to endothelial calcium-activated potassium channel activators.

Throughout, please do not keep redefining abbreviations! The authors must go through their manuscript carefully and delete for example “spontaneously hypertensive rat(s)” the 2nd, 3rd, etc time the full term appears. Only “SHR” should be shown subsequent to its first appearance after the full term. The same applied to “nitric oxide (NO)”, but there are others that need to be fixed as well.

- We apologize for repeatedly defining a given abbreviation throughout the text. In the past, we have been asked to include such definitions in each major section of the manuscript to help improve understanding for the reader. In the present situation, we have followed the reviewer’s suggestion and fully described an abbreviation only upon its first appearance in the text. Subsequent use of the same abbreviation does not include its associated full definition. The only exception is that some terms and associated abbreviations still appear in figure legends in order to facilitate readability and comprehension. We have also strived to use this approach for as many common abbreviations as possible, and have carefully checked the text, as recommended.

Line

24: Authors must learn that they must NEVER use “however” as a conjunction. MUST start a NEW SENTENCE! --- “…. from SHRs. However, responses …”

(The authors should be congratulated for correctly using “however” in line 231.)

- We apologize for this grammatical faux pas, and thank the reviewer for pointing out this error. We have revised the use of “however” throughout the text, as recommended.

25, 342, 368: Change to “i.p.”

- All instances of “I.P.” appearing the text have been revised to “i.p.”, as suggested.

26: There is no such word in the English language as “pharmacologic”. This is dreadful American jargon! Change to “pharmacological”.

- Our apologies for the incorrect word usage. In all instances, “pharmacologic” has been changed to “pharmacological”.

56-66: Delete. This is a summary of the results and must not appear in the Introduction.

- We have omitted the majority of text in question, as recommended. We have only retained the final sentence, which now reads as follows: “Based on our results, we suggest that the SHR model exhibits limited responsiveness to the known vasodilatory actions of SKA-31.”

239: Delete “observed” as it is redundant.

- The text has been revised, as suggested.

243: Once again the use of “however” is incorrect. The word ‘however” may NEVER be used to join two sentences. Must start a new sentence!

- We have revised the text by beginning a new sentence with the use of “however”. The revised text now appears as follows: “Following 2-3 days of rest, administration of SKA-31 (30 mg/kg) to the same SHRs elicited similar spikes in MAP and HR. However, these increases did not readily reverse and were sustained above baseline over the four hour period following injection.”

390: “sec” is a trigonometric term and must never be used as the abbreviation for “seconds”. The SI abbreviation is “s”. Here use “seconds”. In 451 the authors use both “s” and “sec”!?!

- We apologize for the confusion created by this incorrect abbreviation. We have replaced the abbreviation and used the full term “second” in the text. In the second instance, we have inserted the abbreviation “s” for the term “second” throughout the sentence describing individual timing cycles for the PCR protocol.

392: Why “intraperitoneal” here but “i.p.” elsewhere?

- In the revised manuscript, we introduce the term “intraperitoneal” and its abbreviation “i.p.” in Section 2.4 (page 19). Throughout the remainder of the manuscript, only the abbreviation is used. We thank the reviewer for catching this inconsistency.

REFERENCES

These need to be edited so that they conform with journal style. Titles of journals must be capitalized. If journal style requires abbreviated journal names. 

Article titles that are capitalized (e.g., refs 1, 2, 12, 13) must be changed to lower case.

592: Should this journal name include “Scandinavica”?

- We have revised the formatting of references in the bibliography to be consistent with the style recommended by the journal. With respect to the journal title associated with reference #27, Acta Physiologica appears to be the accepted listing, according to the Scandinavian Physiological Society.

Reviewer 2 Report

This is an interesting study investigating the cardiovascular activity of a small molecule opener of KCa2/3 channels in several vascular beds of normotensive rats and hypertensive SHR and in-vivo in SHR. Besides published works by the same group and by others, the study provides some additional insight into whether or not SKA-31 is capable of lowering blood pressure. The results clearly show that the opener is ineffective of lowering blood pressure over several hours but rather increases blood pressure and heart rate in SHR. Control data from Wistar rats would be helpful.

Specific comments:

1. Introduction:

The authors should consider rewrite the last sentences of this section because the current phrasing appears more appropriate for the discussion section.

2. Results:

First paragraph:

The description of the SHR model of genetic primary hypertension could be more specific and may include “sympathetic rush” as one important feature.

3. Figure 1 : The authors state that the constriction to PE is alike in both strain.

Yet, the traces shown suggest a substantial weaker constriction in SHR than in Wistar. The number of experiments (n=3) is rather small. Yet, the responses to SKA-31 and agonists appear clear and confirm this type of endothelial dysfunction in SHR.

4. SKA-31-effects: The traces shown in Figure 2 A and B suggest that there is substantial potentiation of the ACh response by SKA-31 in mesenteric arteries of both strains and in cremaster arteries of the SHR. Perhaps you may select a more representative trace or increase the currently small number of experiments to foster your conclusions. The title and legend to this Figure could better reflect the findings. The sequence of testing the opener and agonists is changed in the trace shown in panel B. Was there any reason to do so or does this affect the responses?

Line 126: Please start with “In myogenically constricted cremaster arteries, …..”

5. Figure 2: Immuno blots

The WB for KCa3.1 does not look very convincing because the size of the protein appears to be too low for a 48 kDa protein and there are several other bands. This is a well-known problem when using currently available ABs against KCa3.1. I am not sure whether this WB analysis of channel protein expression is indeed meaningful. Electrophysiological measurements of protein function may be more appropriate. Please discuss.

Line 230: SKA-313

6. In vivo SKA-31 administration and blood pressure measurements: This part of the manuscript together with the Langendorff-experiments provides insight that is more novel.

In the legend to Figure 6, you write “No statistical difference between SHR and Wistar was noted in the mean data for either MAP or HR obtained pre-injection.” Yet, I could not find the data in the manuscript. Please include the “Wistar” data.

Comment: It seems clear that SKA-31 does not lower, but rather increases MAP and HR over several hours after injection. I am wondering whether there is an early response within minutes after injection. Please revise. In the discussion, you may consider that there could be a SKA-31-provoked desensitization of the targets (KCa2.3/KCa3.1).

7. Discussion: Lines 281-282: You state: “In contrast, bath addition of the KCa channel activator SKA-31 evoked either similar or modestly reduced vasodilatory responses in equivalent vascular tissues from both species.” Do you mean strains?

As I understand the data in Figure 1 and B, there was an increase in the vasodilation in mesenteric arteries of Wistar rats. Please revise.

Line 288: please state coronary “flow” in Wister control rats.

Line 293: “ and in the coronary circulation…” Please skip “intact” as it could be misleading here when dealing with SHR.

Line 295: The term “EDH” may not be appropriate here, as you did not specifically look at EDH.

Line 296: I would be more cautious when comparing SHR with patients. Please revise.

Author Response

Responses to Queries from Reviewer #2

Top of Form

Open Review

English language and style

( ) Extensive editing of English language and style required
( ) Moderate English changes required
(x) English language and style are fine/minor spell check required
( ) I don't feel qualified to judge about the English language and style

Yes

Can be improved

Must be improved

Not applicable

Does the introduction provide   sufficient background and include all relevant references?

( )

(x)

( )

( )

Is the research design   appropriate?

( )

(x)

( )

( )

Are the methods adequately   described?

( )

(x)

( )

( )

Are the results clearly presented?

( )

(x)

( )

( )

Are the conclusions supported by   the results?

( )

(x)

( )

( )

Comments and Suggestions for Authors

This is an interesting study investigating the cardiovascular activity of a small molecule opener of KCa2/3 channels in several vascular beds of normotensive rats and hypertensive SHR and in-vivo in SHR. Besides published works by the same group and by others, the study provides some additional insight into whether or not SKA-31 is capable of lowering blood pressure. The results clearly show that the opener is ineffective of lowering blood pressure over several hours but rather increases blood pressure and heart rate in SHR. Control data from Wistar rats would be helpful.

 We thank the reviewer for his/her constructive comments and overall positive assessment of our submitted manuscript. Our responses to individual concerns are presented below. Note that major revisions to the text are highlighted in yellow to facilitate identification.

Specific comments:

1. Introduction:

The authors should consider rewrite the last sentences of this section because the current phrasing appears more appropriate for the discussion section.

- In response to the reviewer’s comment, we have modified the final sentence of the Introduction section to read as follows: “Based on our results, we suggest that the SHR model of primary hypertension exhibits altered responsiveness to the known vasodilatory actions of SKA-31.”

2. Results:

First paragraph:

The description of the SHR model of genetic primary hypertension could be more specific and may include “sympathetic rush” as one important feature.

- We have improved our description of the SHR model by modifying the text as follows:

“In the present study, we have examined the effects of acute SKA-31 treatment on cardiovascular function in the SHR, an established model of primary hypertension that exhibits cardiac hypertrophy, heart failure, renal dysfunction and impaired regulation of sympathetic outflow (Pinto et al, 1998; Zugck et al, 2013).”

3. Figure 1: The authors state that the constriction to PE is alike in both strain.

Yet, the traces shown suggest a substantial weaker constriction in SHR than in Wistar. The number of experiments (n=3) is rather small. Yet, the responses to SKA-31 and agonists appear clear and confirm this type of endothelial dysfunction in SHR.

- In Section 2.1 of the Results, we describe mean data indicating that PE-induced constriction of small mesenteric arteries pressurized to 70 mmHg is similar in Wistar rats and SHRs (32.3 ± 5.0% and 37.7 ± 2.5%, respectively). In Figure 1A, PE produces a constriction of approximately 130 microns in a Wistar mesenteric artery with a maximal passive intraluminal diameter of approximately 340 microns, leading to a relative reduction in maximal diameter of about 38%. In Figure 1B, the representative SHR mesenteric artery has a maximal passive diameter of approximately 255 microns, and the amplitude of PE-induced contraction is ~105 microns, resulting in a reduction of 43%. Thus, PE does not produce a weaker constriction in the SHR artery, relative to its maximal diameter. As arteries isolated from the same section of the vascular tree of different animals will vary somewhat in diameter, the standard analytical approach is to express the stimulated constriction as a change normalized to the maximal intraluminal diameter of the vessel. Thus, even though the absolute magnitude of the PE-induced constriction in the SHR artery is smaller than that observed in the Wistar mesenteric artery, the relative change in maximal intraluminal diameter is comparable between the two vessels.

4. SKA-31-effects: The traces shown in Figures 1 A/B and 2 A/B suggest that there is substantial potentiation of the ACh response by SKA-31 in mesenteric arteries of both strains and in cremaster arteries of the SHR. Perhaps you may select a more representative trace or increase the currently small number of experiments to foster your conclusions. The title and legend to this Figure could better reflect the findings. The sequence of testing the opener and agonists is changed in the trace shown in panel B. Was there any reason to do so or does this affect the responses?

- The representative tracings displayed in Figure 1A and 1B largely reflect the qualitative pattern of responses observed in our experiments, along with the quantitative data presented in the accompanying histogram (Fig. 1C). A similar rationale was used for the selected tracings displayed in Figure 2A and B. These representative tracings are also relatively “clean”, as a result of the consistent video tracking of intraluminal vessel diameter, and provide clear illustrations of the responses to individual drug treatments. As it is very difficult to select raw tracings that closely reflect the mean responses to all experimental conditions utilized in a given protocol, we are reluctant to replace these tracings with others, as new tracings will also not fully represent the mean data. We are also reluctant to carry out additional experiments solely to obtain tracings that may appear to be more representative, as the number of drug applications and conditions in our experimental protocols increase the likelihood of obtaining one or more spurious responses. Although the number of experiments is small in some instances, we are confident that the pattern of responses to individual applications and conditions are sufficiently robust to allow for meaningful interpretation of the data. Nonetheless, we acknowledge the reviewer’s comment that increasing the number of observations would likely strengthen the data set. As suggested further by the reviewer, we have revised the titles and legends for Figures 1 and 2 in an effort to describe the displayed data more clearly. With respect to the last query, we occasionally change the order of drug application in an experimental protocol to convince ourselves that pharmacological responses to a series of different drug treatments in the same preparation are not influenced by the effects of the preceding drug. In our hands, we typically find that the order of drug applications in a series does not impact the nature of an individual response, providing that washout of the preceding treatment is complete and tissue activity returns to the basal level.

Line 126: Please start with “In myogenically constricted cremaster arteries, …..”

 - We have modified the text as suggested.

5. Figure 2: Immuno blots

The WB for KCa3.1 does not look very convincing because the size of the protein appears to be too low for a 48 kDa protein and there are several other bands. This is a well-known problem when using currently available ABs against KCa3.1. I am not sure whether this WB analysis of channel protein expression is indeed meaningful. Electrophysiological measurements of protein function may be more appropriate. Please discuss.

- We agree that several commercially available antibodies against KCa3.1 channel have not been well characterized, and may not be experimentally reliable. To convince ourselves that the antibody used in our studies is detecting the correct target protein, we carried out western blots using mesenteric arteries isolated from wild-type mice and genetic knockout mice lacking KCa3.1 channel expression. As shown by the preliminary results below (see Supplemental Figure 1), our anti-KCa3.1 channel antibody detects a ~45 kDa band in WT mesenteric arteries under conditions of denaturing SDS-PAGE that is similar in size to recombinant human KCa3.1 channel alpha subunit expressed in HEK293 cells. In arteries from KCa3.1 knockout mice, an equivalent band is not observed. (Please note that the figure below has been included to address the reviewer’s concern, and is part of the supplemental information for another study. Therefore, it cannot be included in the data set for the present manuscript.) In the western blot data displayed in Figure 3E of our manuscript, the predominant band detected by the same anti-KCa3.1 antibody migrated at ~40 kDa and was similar in size to the recombinant human KCa3.1 channel protein used as a positive control. Notably, MacKinnon and colleagues (Lee et al, Science 360: 508, 2018) have recently shown that the alpha subunit of purified human KCa3.1 channel migrates as a ~37 kDa protein upon denaturing SDS-PAGE, which is somewhat smaller than its predicted size of 48 kDa, as the reviewer states. Other investigators have described the KCa3.1 channel subunit as a ~55 kDa band following western blot analysis, so it appears that a range of sizes have been reported. Based on these collective observations, we are convinced that the western blot data displayed in Figure 3 for KCa3.1 channel expression in SHR and Wistar mesenteric arteries are valid and interpretable. In addition, the similar functional responses evoked by the endothelial KCa channel activator SKA-31 in mesenteric, cremaster and coronary arteries from SHR and Wistar rats argue that KCa2.3 and KCa3.1 channel activities are comparable in these vessels and support the interpretations of our western blot data. Although we agree that these results, along with those for the KCa2.3 channel, would be strengthened by the addition of electrophysiological data to demonstrate similar current densities in acutely isolated endothelial cells from SHR and Wistar arteries, we don’t believe that the considerable effort and resources required to undertake these experiments are warranted, in light of our existing data.

Line 230: SKA-313

- We thank the reviewer for pointing out this typographical error; it has been corrected.

6. In vivo SKA-31 administration and blood pressure measurements: This part of the manuscript together with the Langendorff-experiments provides insight that is more novel.

In the legend to Figure 6, you write “No statistical difference between SHR and Wistar was noted in the mean data for either MAP or HR obtained pre-injection.” Yet, I could not find the data in the manuscript. Please include the “Wistar” data.

- The sentence as originally written was incorrect, as we only obtained in vivo radio-telemetry data for SHRs. The original statement has been corrected and appears in the legend as follows: “No statistical difference was observed in the mean data for either MAP or HR obtained pre-injection.” We apologize for the confusion that was created by this oversight in the original manuscript.

Comment: It seems clear that SKA-31 does not lower, but rather increases MAP and HR over several hours after injection. I am wondering whether there is an early response within minutes after injection. Please revise. In the discussion, you may consider that there could be a SKA-31-provoked desensitization of the targets (KCa2.3/KCa3.1).

 - The mean data in Figure 6A provide a hint that SKA-31 administration may induce a modest hypotensive effect within the first 20-30 min, but is difficult to ascertain, due to the injection artefact. In response to a similar comment from Reviewer #3, we speculate that SKA-31 administration may have evoked a modest hypotensive effect that triggered a compensatory elevation in heart rate and presumably cardiac output. Given that SHRs exhibit dysfunction at the levels of the vasculature, the heart and autonomic outflow, it is possible that the prolonged rise in blood pressure may reflect an overcompensation in homeostatic response to a SKA-31 induced reduction in peripheral vascular resistance and mean arterial pressure. We have revised the Discussion by speculating that the observed elevation in blood pressure may be linked to dysfunction in one or more processes contributing to cardiovascular homeostasis associated with an acute hypotensive event. With regards to the reviewer’s second point, we had not considered that an acute bolus injection of SKA-31 may lead to either a desensitization or down-regulation of endothelial KCa channel activities. In larger mammals (i.e. dogs and pigs), it doesn’t appear that repeated I.V. administration of SKA-31 leads to tachyphalaxis, although it is possible that the situation may differ in the SHR. We have included a statement in the Discussion section indicating that the observed elevation in blood pressure may reflect a desensitization of endothelial KCa channel activity and associated reduction in vasodilatory capacity.

7. Discussion: Lines 281-282: You state: “In contrast, bath addition of the KCa channel activator SKA-31 evoked either similar or modestly reduced vasodilatory responses in equivalent vascular tissues from both species.” Do you mean strains?

- We agree with the reviewer’s comment and have changed “species” to “strain” in the identified text.

As I understand the data in Figure 1 and B, there was an increase in the vasodilation in mesenteric arteries of Wistar rats. Please revise.

- Without reference to a specific line #, we have assumed that the reviewer is referring to our description of the dilatory effects observed in the presence of a threshold concentration of SKA-31, originally presented on lines 287-289. We have revised this statement by adding the following sentence: “SKA-31 treatment also augmented ACh-mediated dilation in mesenteric arteries from Wistar rats and SHRs (Fig. 1C).”

Line 288: please state coronary “flow” in Wister control rats.

- We have modified the sentence, as recommended.

Line 293: “ and in the coronary circulation…” Please skip “intact” as it could be misleading here when dealing with SHR.

- We have modified the text, as recommended.

Line 295: The term “EDH” may not be appropriate here, as you did not specifically look at EDH.

- In response to the reviewer’s comment, we have changed the term “EDH” to “endothelium-mediated dilation” in the identified section of text.

Line 296: I would be more cautious when comparing SHR with patients. Please revise.

- We agree that a more cautious statement would be prudent. We have revised the sentence in question to read as follows: “The widespread nature of this impairment in SHR vasculature is reminiscent of observations in patients showing that coronary endothelial dysfunction co-associated with similar dysfunction in brachial arteries.

Supplementary Figure 1 - Detection of KCa3.1 protein in mesenteric arteries from WT and KCa3.1-/- mice. (A) Western blot showing immuno-reactive bands detected with an anti-KCa3.1 channel antibody (clone D5, Santa Cruz Biotechnology) in mesenteric arteries from 3 individual wild-type (WT) and KCa3.1 knockout (-/-) mice. HEK 293 cells transfected with cDNA encoding either human KCa3.1 channel or rat Kir2.1 channel were used as positive and negative staining controls, respectively. KCa3.1 channel protein is denoted by the arrow on the left-hand side. Detection of β-actin was used as a loading control. (B) Histogram quantifying the protein expression ratio of KCa3.1 to β-actin in mesenteric artery homogenates from WT and KCa3.1 -/- mice, along with lysates from transfected HEK 293 cells. The asterisk signifies a statistical difference (P < 0.05) between the indicated groups, as determined by ANOVA and a Tukey post-hoc test, (WT, n = 4 and KCa3.1-/- , n = 6 animals).

Reviewer 3 Report

Major points

1. How is the expression of BK receptors in the experiment 5? If the the author's hypothesis that the vasodilation by BK in the presence of SKA-31 depends on the expression of BK receptors (Fig.3) is true, maybe it would be elevated.

2. Similarly, how is the expression of Ach receptors in the fig 3?

3. How do you explain why MAP is elevated by the administration of SKA-31 (Fig.6)

Minor points

1. In the figure 1C/2C, explanation of which square denotes Wistar/SHR as in the figure 5C would help readers to understand the figure.

2. line 195---typo??

Author Response

Responses to Queries from Reviewer #3

Open Review

English language and style

( ) Extensive editing of English language and style required
( ) Moderate English changes required
(x) English language and style are fine/minor spell check required
( ) I don't feel qualified to judge about the English language and style

Yes

Can be improved

Must be improved

Not applicable

Does the introduction provide   sufficient background and include all relevant references?

(x)

( )

( )

( )

Is the research design   appropriate?

( )

( )

(x)

( )

Are the methods adequately   described?

(x)

( )

( )

( )

Are the results clearly presented?

( )

(x)

( )

( )

Are the conclusions supported by   the results?

( )

( )

(x)

( )

Comments and Suggestions for Authors

 We thank the reviewer for his/her constructive comments and overall positive assessment of our submitted manuscript. Our responses to individual concerns are presented below. Note that major revisions to the text are highlighted in yellow to facilitate identification.

Major points

1. How is the expression of BK receptors in the experiment 5? If the author's hypothesis that the vasodilation by BK in the presence of SKA-31 depends on the expression of BK receptors (Fig.3) is true, maybe it would be elevated.

- We have assumed that the reviewer is referring to Figure 5 of our manuscript, which quantifies stimulated changes in coronary flow, left ventricular developed pressure and heart rate in isolated hearts from SHRs and Wistar rats. We did not directly assess BK receptor expression in the coronary vasculature, as these vessels are very difficult to isolate from rodent myocardium (they are typically embedded in the tissue and not surface-exposed, as seen in larger mammals). Our preliminary efforts towards this goal indicated that we could not isolate a sufficient number of coronary arteries to perform western blots in a reliable manner. Carrying out qPCR analysis alone for BK receptor expression in coronary arteries was deemed inadequate, as there are numerous examples in the literature and in our own studies demonstrating that changes in mRNA level do not reliably predict changes in protein expression for a given target. As a result of these technical difficulties, we have been unable to directly quantify BK receptor expression in the coronary vasculature of SHR and Wistar rat hearts.

With regards to the second point, we have speculated that the reduced expression of endothelial BK receptors observed in small arteries from the mesenteric bed could explain the lower vasodilatory responses to BK in these vessels, and that a similar situation may also explain the absent/reduced responses to BK in arteries from cremaster skeletal muscle and coronary vasculature. If endothelial BK receptor signal transduction and associated calcium mobilization were also compromised in the SHR, then we would anticipate that SKA-31-mediated augmentation of stimulated dilation would be lower, or perhaps absent. The reviewer’s suggestion that reduced BK receptor expression and evoked dilation may allow for an elevated enhancement of the primary response by SKA-31 is interesting, and a point we did not consider. In order for this scenario to occur, agonist-evoked calcium elevations in the endothelium would need to be of sufficient magnitude to stimulate SKA-31 sensitized KCa channels, and these channels would need to be correctly localized to sense such elevations. In the SHR coronary circulation, it is possible that endothelial BK receptor activation does not elevate cytosolic calcium sufficiently to activate even SKA-31 sensitized KCa channels, or that these channels cannot sense the elevation, due to altered cellular distribution. We have added a sentence to the Discussion section to highlight this possibility.

2. Similarly, how is the expression of Ach receptors in the fig 3?

- Vasodilatory responses to acetylcholine (ACh) in mesenteric and cremaster skeletal muscle arteries were used primarily as an internal control to examine the extent of endothelial dysfunction in SHRs and provide a comparison with vasoactive responses to BK. It is well established that ACh-evoked dilation is typically compromised in the setting of endothelial dysfunction, which can serve as a useful functional biomarker. Examining muscarinic receptor expression, particularly the M3 isoform, in whole arteries is difficult, as this receptor isoform is present in both vascular endothelium and smooth muscle. Interpretation of such data would therefore be challenging. For these reasons, we did not explicitly examine M3 muscarinic receptor expression in mesenteric arteries.

3. How do you explain why MAP is elevated by the administration of SKA-31 (Fig.6)

- We are also puzzled by this observation, but the in vivo telemetry data may provide a clue. As shown in Figure 6B, acute SKA-31 administration leads to an elevation in heart rate that would be expected to increase cardiac output (assuming that venous return is not altered). If the putative increase in cardiac output overcompensated for a modest reduction in systemic vascular resistance induced by SKA-31, then mean arterial pressure would rise. In Figure 6A, there is a hint that SKA-31 administration decreased MAP within the first 30 min following injection, whereas a similar drop was not present in SHRs injected with drug vehicle alone. It is thus possible that acute SKA-31 administration produced an initial reduction in vascular resistance and MAP that triggered a compensatory increase in heart rate and cardiac output that overcame the hypotensive action of SKA-31. If the compensatory homeostatic response to SKA-31 administration were exaggerated, as a result of dysfunction in the vasculature, heart or autonomic control of these systems, then an overshoot in blood pressure may occur. Even though this scenario remains speculative, a statement has been added to the Discussion to highlight such a possibility.

Minor points

1. In the figure 1C/2C, explanation of which square denotes Wistar/SHR as in the figure 5C would help readers to understand the figure.

- We agree with the reviewer that the recommended change should improve clarity of Figures 1 and 2. Changes to the symbol legend in panel C of each Figure have been made, as suggested.

2. line 195---typo??

- It appears that some of the text in our original manuscript was reformatted during the processing of the draft by the journal’s editorial system. The sentence appearing on line 195 of the reviewers’ copy should read as follows:

Bolus injections of saline or drug vehicle (i.e. equivalent volumes) had no effects on total coronary flow, LV developed pressure or heart rate in either preparation (Figure 5A-C).”

A quick search of the processed manuscript indicated that the missing part of the sentence incorrectly appeared at the end of the legend for Figure 5. This error has now been corrected in the revised manuscript.

Round 2

Reviewer 2 Report

I have no further comments.